# Multi-center prospective population pharmacokinetic study and the performance of web-based individual dose optimization application of intravenous vancomycin for adults in Hong Kong: A study protocol

Ka Ho Matthew Hui[1], Chung Yan Grace Lui[2], Ka Lun Alan Wu[3], Jason Chen[4], Yin Ting Cheung[1]*, Tai Ning Teddy Lam[1]*

1 School of Pharmacy, Faculty of Medicine, The Chinese University of Hong Kong, Hong Kong, China, 2 Department of Medicine and Therapeutics, Faculty of Medicine, The Chinese University of Hong Kong, Hong Kong, China, 3 Department of Clinical Pathology, Pamela Youde Nethersole Eastern Hospital, Hong Kong East Cluster, Hospital Authority of Hong Kong, Hong Kong, China, 4 Department of Pharmacy, Ruttonjee and Tang Shiu Kin Hospitals, Hong Kong East Cluster, Hospital Authority, Hong Kong, China

* teddylam@cuhk.edu.hk (TNTL); yinting.cheung@cuhk.edu.hk (YTC)

**Data Availability Statement:** No datasets were generated or analysed during the current study. All

## Abstract

A recent consensus guideline recommends migrating the therapeutic drug monitoring practice for intravenous vancomycin for the treatment of methicillin-resistant *Staphylococcus aureus* infection from the traditional trough-based approach to the Bayesian approach based on area under curve to improve clinical outcomes. To support the implementation of the new strategy for hospitals under Hospital Authority, Hong Kong, this study is being proposed to (1) estimate and validate a population pharmacokinetic model of intravenous vancomycin for local adults, (2) develop a web-based individual dose optimization application for clinical use, and (3) evaluate the performance of the application by comparing the treatment outcomes and clinical satisfaction against the traditional approach. 300 adult subjects prescribed with intravenous vancomycin and not on renal replacement therapy will be recruited for population pharmacokinetic model development and validation. Sex, age, body weight, serum creatinine level, intravenous vancomycin dosing records, serum vancomycin concentrations *etc.* will be collected from several electronic health record systems maintained by Hospital Authority. Parameter estimation will be performed using non-linear mixed-effect modeling techniques. The web-based individual dose optimization application is based on a previously reported application and is built using R and the package *shiny*. Data from another 50 subjects will be collected during the last three months of the study period and treated as informed by the developed application and compared against historical control for clinical outcomes. Since the study will incur extra blood-taking procedures from patients, informed consent is required. Other than that, recruited subjects should receive medical treatments as usual. Identifiable patient data will be available only to site investigators and clinicians in each hospital. The study protocol and informed consent forms have been approved by the Joint Chinese University of Hong Kong–New Territories East

relevant data from this study will be made available upon study completion.

**Funding:** The author(s) received no specific funding for this work.

**Competing interests:** The authors have declared that no competing interests exist.

Cluster Clinical Research Ethics Committee (reference number: NTEC-2021-0215) and registered at the Chinese Clinical Trial Registry (registration number: ChiCTR2100048714).

## Introduction

### Indication of intravenous vancomycin

Intravenous (IV) vancomycin has long been the first-line antimicrobial agent for treating severe methicillin-resistant *Staphylococcus aureus* (MRSA) infection [1]. It is often prescribed empirically for patients with suspected MRSA infection because of its efficacy in organism eradication. However, a significant drawback of IV vancomycin is the risk of acute kidney injury (AKI), which is associated with large areas under the curve of the concentration-time profile of vancomycin [2]. Given the narrow therapeutic index of IV vancomycin, therapeutic drug monitoring (TDM) of vancomycin is mandatory to balance efficacy against toxicity [3].

### Traditional TDM of IV vancomycin

In 2009, the American Society of Health-System Pharmacists, the Infectious Diseases Society of America, and the Society of Infectious Diseases Pharmacists jointly published a consensus report (the 2009 guideline), recommending a steady-state area under the curve over 24 hours ($AUC_{24}$) to minimum inhibitory concentration (MIC) ratio ($AUC_{24}$/MIC) of at least 400 mg·hr/L as the pharmacokinetic (PK) target for successful vancomycin therapy. Nevertheless, given the difficulty in obtaining multiple serum vancomycin concentrations ($C_s$) for the estimation of $AUC_{24}$, they recommended the use of steady-state trough $C_s$ as a surrogate marker for the $AUC_{24}$/MIC target [4].

### Updated recommendations for TDM of IV vancomycin

Despite the previous recommendation, over the past decade, there has been minimal to no data on the efficacy and safety profile supporting the use of steady-state trough $C_s$ as a helpful treatment endpoint [5]. In fact, steady-state trough $C_s$ is later found to be poorly correlated with $AUC_{24}$ [6]. Contrastingly, there has been increasing evidence on the use of $AUC_{24}$ as the therapeutic target for IV vancomycin [7, 8]. Moreover, on top of the previous target of $AUC_{24}$/MIC of at least 400 mg·hr/L to eradicate MRSA, it has also been shown that $AUC_{24}$ of at least 600 mg·hr/L is associated with an increased risk of AKI [9]. Of equal importance, Neely et al. conducted a prospective trial demonstrating the superiority of combining the $AUC_{24}$ target and a Bayesian approach to $AUC_{24}$ estimation over the traditional trough-based approach [8]. In fact, Bayesian tools are becoming more readily available over the past decade, thanks to rapid advancement in computing efficiency [10–13].

Given the above, the above named societies and the Pediatric Infectious Diseases Society published a revised consensus guideline in 2020 (the 2020 guideline). It recommended (1) *against* the previous therapeutic target in terms of steady-state trough $C_s$, and (2) the promotion of the achievement of the therapeutic target of $AUC_{24}$ between 400–600 (assuming MIC of 1 mg/L) through dose individualization guided by individual AUC24, which should be obtained through Bayesian estimation based on trough Cs (or both trough $C_s$ and peak Cs for better accuracy) [5].

## Advantages of Bayesian estimation

Since trough-based TDM is based on observations at the steady state, it is often necessary to wait until the third or fourth infusion, after which the steady state is almost reached, for the sampling of Cs. However, with Bayesian estimation, it becomes possible to extrapolate the pre-steady-state PK profile to predict steady-state behavior, and Cs sampled within the first 24 to 48 hours can be sufficient to inform the optimal dosing regimen [5]. It implies that waiting until the steady state for sampling is no longer necessary. As a result, the Bayesian approach can help reduce the sampling of vancomycin levels and shorten the length of therapy [8].

Performing Bayesian estimation requires a prior distribution, which, as recommended by the 2020 guideline and in the current study, will be represented by a population pharmacokinetic (popPK) model of IV vancomycin among adults in Hong Kong [5]. The popPK model contains the population estimates of the effects of multiple covariates, including but not limited to age, sex, body weight (WT), serum creatinine level (SCr), and distributions. Compared to the traditional trough-based approach, which is based solely on the observed Cs, the currently recommended Bayesian approach also takes the population characteristics into account, thus improving the accuracy of the estimation of $AUC_{24}$ [14].

Besides, since Bayesian estimation requires numerical approximation and cannot be performed manually, a web-based application that automates the estimation and dose optimization process would be ideal for clinical use. Such an interface will be designed to automatically obtain individual medical data from electronic health record systems, thus reducing the workload of the clinical frontlines and the incidence of medication errors.

## Current practice in Hospital Authority and recommendations

The TDM practice of IV vancomycin in medical institutions under Hospital Authority (HA) in Hong Kong has generally been in line with the 2009 guideline that steady-state trough $C_s$ is being monitored for patients put on IV vancomycin. In view of the updated evidence and guideline, we recommend local institutions to advance the current practice to meet the suggestions by the 2020 guideline to improve the treatment outcomes. In support of the implementation of the dose optimization application, the current study is being proposed to (1) estimate and validate a popPK model of IV vancomycin for adults in Hong Kong, (2) develop a web-based dose optimization application for clinical use in HA, and finally (3) evaluate the performance of the web-based dose optimization application by comparing the treatment outcomes and clinical satisfaction against the traditional trough-based TDM approach.

## Materials and methods

### Study design and population

This is a multi-center prospective study involving hospitals across all seven clusters (which corresponding to different regions of Hong Kong) of HA. All in-patient subjects (1) at least 18 years of age, (2) admitted to one of the following nine HA hospitals: Pamela Youde Nethersole Eastern Hospital, Ruttonjee and Tang Shiu Kin Hospitals, Queen Mary Hospital, Queen Elizabeth Hospital, Kwong Wah Hospital, United Christian Hospital, Princess Margaret Hospital, Prince of Wales Hospital, and Tuen Mun Hospital, and (3) for whom intermittent IV vancomycin is prescribed are eligible for recruitment into the study for data collection. Notwithstanding the above, all subjects prescribed any form of renal replacement therapy that has been started before or is scheduled to start during IV vancomycin treatment shall be excluded.

This study expects to recruit 300 subjects, among which 50 subjects will be set aside for external validation of the popPK model, while the rest belong to the model estimating set. At a

| | Enrolment | | Post-enrolment | | | |
|---|---|---|---|---|---|---|
| TIMEPOINT | Day -1[a] | Day 0 | Day 0 | Day 1 | Day2 | Afterwards[b] |
| **ENROLMENT** | | | | | | |
| *Eligibility screen* | (X) | X | | | | |
| *Informed consent* | | X | | | | |
| **INTERVENTIONS** | | | | | | |
| *Vancomycin treatment* | | | ◆━━━━━━━━ | | ◆ – – – – ◆ | |
| **ASSESSMENTS** | | | | | | |
| *Constant data items* | | | X | | | |
| *Dosing records* | | | X | X | X | (X) |
| *Vancomycin concentrations* | | | X | X | X | (X) |
| *Other dynamic data items* | | | X | X | X | (X) |

**Fig 1. The SPIRIT schedule of enrolment.** [a]Informed consent may be obtained on the next day if it cannot be done on the day the subject is screened. [b]Collection of dynamic data items will continue until vancomycin treatment ends.

later stage of the study, data from another 50 subjects will be collected for the evaluation of the performance of the developed interface. Multi-center involvement is expected to facilitate adequate enrolment.

## Expected study timeline

The entire data collection period will last for one year. It is expected that the popPK data collection period (PDCP) will last for about six months. popPK model development will start once sufficient data have been collected for preliminary analyses. Background preparation work for the dose optimization interface will be in progress during the entire popPK study period. After the interface becomes ready-to-use, training to use the interface and evaluation of the performance of the developed interface are expected to start at about 9 months into the study. The last 3 months of the study period is the evaluation data collection period (EDCP). The SPIRIT schedule of enrolment is available in Figs 1 and 2 outlines the study timeline.

## Data items to collect

**Data item summary.** For each subject, constant data items refer to variables that take only a single value. Meanwhile, all dynamic data items should be collected during each of his/her individual data collection period(s) (IDCP), defined as the period starting from *the start of the 1st infusion* and until *the sampling for the last Cs measurement*. Data items required for the study are summarized in Table 1.

For dynamic data items, all records available during the IDCP of a subject will be taken. The most recent Cs measurement will be considered the last one when (1) the subject has recovered and IV vancomycin treatment has been stopped, (2) the subject has been put off IV vancomycin and switched to receive alternative antibiotic(s), (3) the clinician has decided that IV vancomycin is to be halted indefinitely, or (4) the subject has deceased. (Note that either a temporary halt of IV vancomycin due to, for *e.g.*, impaired renal function, high Cs *etc.*, or a change in IV vancomycin dosing regimen will *not* be considered an interruption of the IDCP.) If, within the PDCP/EDCP, IV vancomycin is started on a subject *again* after his last IDCP, a new IDCP should be initiated for the same subject. If the PDCP/EDCP ends before an IDCP of a subject, data collection for the subject should be extended until the IDCP ends as defined above. Besides, the dates and times of all dynamic data item records should be noted.

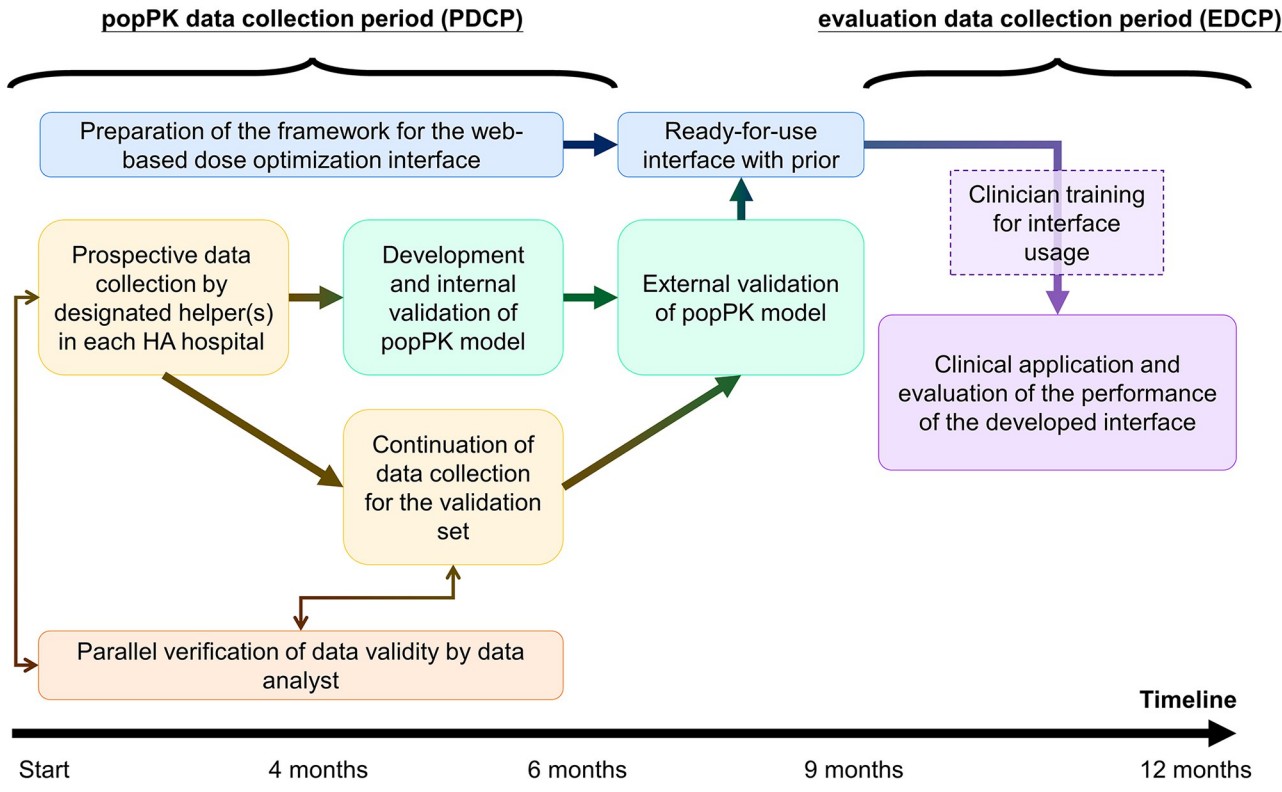

**Fig 2. The study timeline.** popPK: population pharmacokinetic. HA: Hospital Authority.

**Details for IV vancomycin dosing records.** For each infusion, (1) the date and **exact** timing (*accurate to the minute* as much as possible) at the start of infusion, (2) the infusion period assuming constant-rate infusion (with timed records of any irregularity in infusion rate and interruption), and (3) the total amount infused will be recorded.

In the case that it comes to the investigators' attention that the recruited subject either (1) has received IV vancomycin within 72 hours before the start of the 1[st] infusion or (2) has severe baseline renal impairment and has received IV vancomycin within a week before the start of the 1[st] infusion, then all available dosing records within these stated periods prior to the 1[st] infusion should be recorded.

**Details for vancomycin concentration records.** In this paragraph, dosing interval 1 refers to one of the pre-steady-state dosing intervals and dosing interval 2 refers to one of the steady-state or near-steady-state dosing intervals. Subject to the actual implementation in individual hospital level, the recommended sampling schedule requests six $C_s$ measurements. These include (1) *two* peak $C_s$ sampled *at least one* hour and *within two* hours after the last infusion ends during dosing interval 1 and dosing interval 2, (2) *two* random levels sampled *at least* an hour after the last peak $C_s$ and *at least* an hour before the next trough $C_s$ during dosing interval 1 and dosing interval 2, and (3) *two* trough $C_s$ sampled within *one* hour and *strictly* before the start of the *next* infusion during dosing interval 1 and dosing interval 2. In the case that not all the six recommended $C_s$ measurements can be performed (*e.g.* due to failure to obtain informed consent or difficulties in logistics), an alternative sparser schedule should be considered. Details regarding the $C_s$ sampling schedule are available in Fig 3.

In case it is infeasible to sample during a planned dosing interval, sampling should be delayed to the next feasible dosing interval. If vancomycin is put off before the last dose

**Table 1. Data items to be collected.**

| Item type | | Data item |
|---|---|---|
| Constant data items (one value per subject) | | Date of birth |
| | | Ethnicity |
| | | Hospital and ward/specialty |
| | | Sex |
| | | Baseline WT |
| | | Body height |
| | | Obesity status |
| | | Baseline SCr |
| | | Source of infection |
| Dynamic data items | Measurements, assessments, or events | Significant changes in WT |
| | | SCr during treatment |
| | | Pathophysiological conditions: AKI, sepsis, severe trauma, severe burns *etc.* |
| | | Death (if applicable) |
| | | Concomitant drugs (diuretics, aminoglycosides, non-steroidal anti-inflammatory drugs, or other drugs that significantly influence renal functions) |
| | | Renal replacement therapy (intermittent or continuous) |
| | Vancomycin dosing records | Date and exact timing of start and end of infusion |
| | | Infusion rate |
| | | Total amount infused |
| | Cs | Date and exact timing of sampling |
| | | Measured concentrations |
| | Microbial cultures | Date and time of sampling |
| | | Isolated micro-organisms |
| | | MIC of vancomycin against MRSA |
| | outcomes | Time to achieving therapeutic target |
| | | Time to development of AKI at different stages |
| | | Length of IV vancomycin therapy |
| | | Number of $C_s$ blood samples collected |
| | | Time to recovery in terms of time to afebrile |
| | | Time to normal white blood cell count |

AKI: acute kidney injury. Cs: serum vancomycin concentration. IV: intravenous. MIC: minimum inhibitory concentration. MRSA: methicillin-resistant Staphylococcus aureus. SCr: Serum creatinine concentration. WT: body weight. The reference method to measure MIC is broth microdilution but Etest is used in the involved clinics. Previous results have shown comparable measurements between broth microdilution and Etest [15].

intended within the IDCP, then if vancomycin is restarted later, the schedule should be restarted from dosing interval 1. Note that any change in vancomycin dose and/or any supplementary vancomycin dose administered at once should not interrupt the planned sampling schedule. If there is a change in the administration frequency, the sampling schedule should be updated based on the new frequency. The timings of all samplings of Cs after satisfying the above recommended measurement schedule are not bound by the study protocol but subject entirely to clinical needs as judged by the clinicians.

## Data collection method

Clinicians will order extra laboratory assays according to the data items required by the study protocol. Site investigators will collect and organize required data of recruited patient using the Clinical Management System, Medication Administration Record, In-Patient Medication

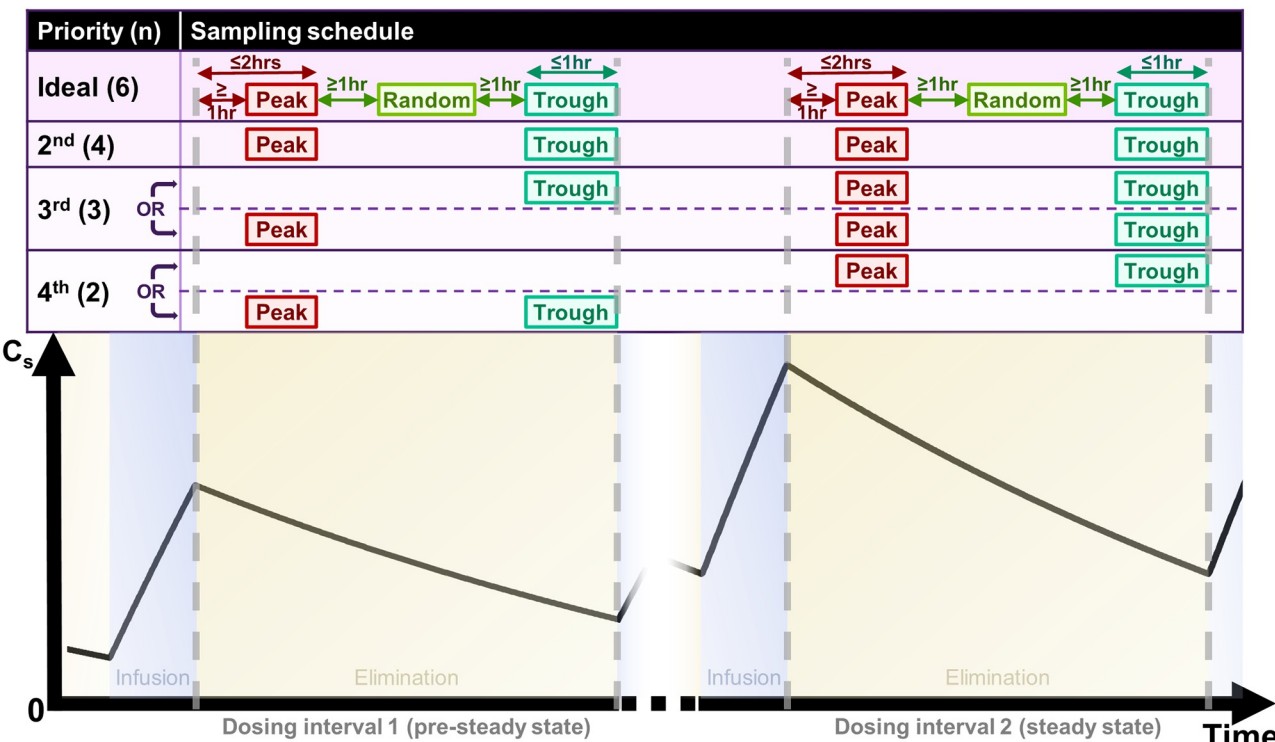

**Fig 3. Details of the $C_s$ sampling schedule with respect to the dosing schedule.** $C_s$: serum vancomycin concentration. n: number of $C_s$ measurements in the schedule. Note that the time separation requirements for peak, random, and trough $C_s$ are only shown in the ideal sampling schedule. The same set of requirements applies for all proposed schedules with lower priorities.

Order Entry System, bedside patient chart, and, when necessary, verbal clarifications. Anonymized patient data will be shared in a confidential manner with the data analysts. The site investigators shall keep a separate, confidential conversion list between anonymized identifiers (available to the data analysts) and in-site patient identifiers, such that clarification will be possible when data validity is in doubt.

### Data exclusion and management

All data in any IDCP (1) with any missing constant data item (except body height unless subject is obese, and ethnicity), (2) without baseline SCr, or (3) without at least one peak $C_s$ (or random $C_s$) plus at least one trough $C_s$ sampled will be removed from subsequent analyses. Besides, in each IDCP, all $C_s$ records that are over 168 hours (7 days) after the last Cs record or the end of the last infusion (whichever later) will be removed from subsequent analyses. All subjects with multiple IDCPs will be treated as the same subject and the vancomycin estimated to be remaining in the subject's body from the previous IDCP will be carried over for the estimation of $C_s$ in the subsequent IDCP(s).

Each of WT, SCr, and body height will be imputed across each IDCP with linear interpolation and extrapolation (or assumed constant if only one value is present). Preliminary analyses of individual Cs profiles will be performed to identify potential outliers. When necessary, investigations into the possibilities of errors in data collection and unusual patient conditions causing the extreme values will be carried out as soon as possible. Besides, regarding the severity of AKI, KDIGO staging will be determined for each subject from the SCr collected (No AKI, stage 1, 2 or 3).

## PopPK model estimation and validation

**Structural and parameter models.**    To characterize the $C_s$-time profiles of known doses and administration times of IV vancomycin, the one- or two-compartment infusion model with first-order elimination will be tested, where the one demonstrating better goodness-of-fit (demonstrated by smaller objective function value) will be chosen. Between-subject and between-occasion variabilities in PK parameters are assumed to follow log-normal distributions. Residual unexplained variability will be described by a combined proportional-additive error model [16–18].

**Covariate model.**    The effects of WT on PK parameters will be presumably estimated by the power model, where allometric scaling with pre-determined exponents will be tested against estimated exponents [19]. Creatinine clearance will be approximated by the Cockcroft-Gault equation (which is based on sex, age, WT, and SCr) and associated with vancomycin clearance by testing different curve functions [20]. Residual covariate effects are then tested against other potential covariates, including but not limited to ethnicity, concurrent pathophysiological conditions, concomitant drugs, renal replacement therapy, and isolated microbes. Hypothesis testing at $\alpha$ = 0.01 will be conducted to compare the goodness-of-fits between two nested models by assuming that the change in objective function value from the richer model to the sub-model follows the $\chi^2$-distribution with df equal to the number of constrained parameters [21].

**Model evaluation and validation.**    Predictive plots, residual plots, normalized empirical Bayes estimates plots, prediction-corrected visual predictive check, and normalized prediction distribution errors will be inspected to evaluate the final model and parameter estimates [22]. After the above evaluation, bootstrapping using 1,000 resamples will be done for internal validation [23]. External validation will be done by evaluating the internally validated model against a separate, smaller dataset.

**Computer software.**    NONMEM® 7 will be used to obtain parameter estimates [24]. Below-limit-of-quantification data will be assessed using the M3 method [25]. Perl-speaks-NONMEM will be used to coordinate NONMEM® runs and model evaluation [26]. R and its packages will be used for the generation of model evaluation graphics [27, 28].

**Expected outcomes.**    In predictive plots, observations should scatter around the identity line. Weighted residuals and normalized empirical Bayes estimates should scatter around zero (with 95% of the points lying within -1.96 and 1.96) with no observable trend alone and against time and other variables. Visual predictive checks should demonstrate agreements between corrected observed and predicted Cs in terms of the percentiles. Normalized prediction distribution error plots should resemble the standard normal distribution. Bootstrapping should show reasonable distributions of bootstrap estimates with their medians close to and their 95% confidence intervals containing the model estimates. Apart from bootstrapping, the above applies also to external validation. Overall, all model diagnostics should indicate good predictive performance and stability of the developed popPK model. Successful results in this part will provide the foundation to conduct Bayesian estimation that can improve the TDM practice of IV vancomycin, as agreed by the community [5].

## Development of the web-based dose optimization application

The infrastructure and framework of a previously published web-based individual dose adjustment application for high-dose methotrexate in the pediatric population will be replicated in this study. This application was built using R and its packages, including *shiny*, which are (1) open sources, (2) validated against the proprietary software package, NONMEM®, for the accuracy of individual parameter estimation, and (3) has been shown to be more efficient than

relying on NONMEM® in performing individual parameter estimation [29]. The interface will be amended to adapt to the requirements of this study and clinical application.

## Statistical analyses for outcomes

The group of subjects treated as informed by the developed application (recruited during the EDCP) is compared against the group treated with the traditional approach (recruited during the PDCP) and historical control data, which will be verified by baseline characteristics comparison and, if necessary, propensity score matching. The probabilities of developing AKI and proportions of AKI stages between approaches will be compared with the chi-square test. The times to achieving therapeutic target, development of AKI at different stages, becoming afebrile, and achieving normal white blood cell count will be compared using Kaplan-Meier analysis and Cox proportional hazard regression. Finally, the length of IV vancomycin therapy and the number of $C_s$ blood samples collected will be compared using the Student's t-test or the Wilcoxon signed rank test, depending on the distribution of the data.

## Ethical considerations

**Need of patient data collection.**   The current trough-based TDM approach of vancomycin is no more recommended due to its poor prediction of vancomycin exposure, and therefore should be phased out. Instead, the currently recommended TDM strategy is to rely on Bayesian estimation of vancomycin $AUC_{24}$. The application of the strategy requires a prior distribution for the local population that is developed upon rich popPK data, thus patient data collection is indispensable. The result of this study will enable HA institutions to comply with the most updated recommendations and is expected to improve treatment outcome once applied.

**Changes to clinical procedures.**   From the perspective of the patients recruited into this study, the study sampling schedule requires more measurements of Cs than usual and thus will likely incur more frequent blood sampling than usual. Other than that, there is no indication of other procedures and medical treatments in this study. As in routine clinical practice, study subjects will receive medical treatments that are deemed the most appropriate by clinicians. It is well acknowledged that rich data collected in study subjects may alter clinicians' decisions on vancomycin treatments. However, when compared to the current trough-based TDM approach, the rich sampling scheme will likely enrich the information required for making accurate prediction of vancomycin exposure. This is supported by the 2020 guideline's recommendation to estimate $AUC_{24}$ using both steady-state peak $C_s$ and steady-state trough $C_s$ when a Bayesian tool is not yet available [5]. Therefore, it is very unlikely that the extra sampling required in this study will adversely affect the optimality of vancomycin therapy.

From the perspective of the clinicians and site investigators, workload may increase due to more frequent sampling, ordering of assays for extra samples, and collection and validation of detailed patient data. However, TDM has been a routine procedure of vancomycin treatment, so new types of intervention are not introduced in this study. In any case, clinicians and site investigators should prioritize patient care over the consolidation of study data.

**Patient privacy.**   Access to identifiable patient data collected in this study will be available only to site investigators and clinicians in each institution. All data will be anonymized before being sent to the data analysts. Data collected by the data analysts will be handled with encryption and password protection.

**Informed consent.**   The gathering of routinely collected data is not expected to adversely affect patient treatment nor expose patient information. However, since rich sampling will incur extra blood-taking procedures done on patients, a written informed consent must be

obtained from the patient or, if the patient is incapable of giving informed consent, his legal representative, who must be a member of his next of kin. Extra blood taking procedures will not be performed for subjects who refuse to participate in the study, and the data available through routine vancomycin treatment based on the updated guideline will continue to be included in this study.

**Compliance to ethical standard, registration, and approval.** This study will be conducted in compliance with the Declaration of Helsinki and is being submitted for review by multiple Cluster Research Ethics Committees of HA. The study protocol (version 1.1, Apr 21$^{st}$, 2021) and informed consent forms have been approved by the Joint Chinese University of Hong Kong–New Territories East Cluster Clinical Research Ethics Committee (reference number: 2021.175; approval date: May 21$^{st}$, 2021). (See S1 File.) Approvals by the Ethics Committees of other clusters are pending. Any protocol modifications will be reviewed by the Ethics Committee. The study has also been registered at the Chinese Clinical Trial Registry (registration number: ChiCTR2100048714) on July 13$^{th}$, 2021. To ensure homogenous study procedures across study centers, the principal investigator and his team of other investigators at The Chinese University of Hong Kong will explain the study procedures thoroughly to the site coordinators and answer all questions they have before study commencement, as well as closely monitor the recruitment of and study procedures done on the first subject at each center.

## Supporting information

**S1 File. Documents approved by the ethics committee.**
(DOCX)

**S1 Checklist. SPIRIT 2013 checklist: Recommended items to address in a clinical trial protocol and related documents** *.
(DOCX)

**S1 Data.**
(PDF)

**S2 Data.**
(PDF)

**S3 Data.**
(PDF)

**S4 Data.**
(PDF)

## Acknowledgments

The authors would like to thank the following individuals for their assistances throughout the preparation and execution of the study (in alphabetical order under each section):

### Site coordinators

Dr. CHIK Thomas Shiu Hong, Dr. FUNG Ka Kin, Dr. HUNG Ling Lung, Dr. LIU Wai To Raymond, Ms. NG Tsz Ming, Dr. SIN Ching Tai Eugene, Dr. TSANG Lok Man.

## Co-investigators

Dr. CHAN Man Chun, Dr. CHAN Shuk Ying, Dr. CHAN Wai, Dr. CHEN Pak Lam Sammy, Mr. CHIU Kenneth Ting Hei, Dr. CHOI Yau Chung, Dr. CHU Man Yee, Ms. CHUNG Ho Man Melissa, Ms. FAN Sheung Yin, Mr. FONG Ming Kit, Mr. KWAN Him Shek, Mr. KWOK Wing To, Dr. KWONG Tsz Shan, Ms. LAI Sin Man Selma, Dr. LAM Kwok Wai, Dr. LAU Pui Ling, Dr. LAU Wing Tung, Ms. LAW Lok Ting, Ms. LEE Chi Po Janet, Dr. LEE Kin Ping May, Dr. LEUNG Wai Sing, Dr. LI Chun Man Timothy, Mr. LI Chun Wai, Prof. LING Ka Kin Samuel, Mr. MAK Wai Ming Raymond, Mr. MAK Wun Cheung, Ms. SO Wing Tung, Dr. TANG Hing Cheung, Dr. TING Wan Man, Dr. TO Pui Yee, Ms. TSANG Chui Shan Rikki, Dr. TSANG Tak Yin Owen, Ms. WONG Ho Yi Angie, Dr. WONG Sin Yue, Ms. WONG Ting Yuk, Prof. WONG Wai Tat, Dr. WU Tak Chiu, Dr. YEUNG Wai Tak Alwin, Dr. YUNG Sai Kwong.

## Heads of involved departments

Dr. CHAN Chak Lam Alexander, Dr. CHAN Ngai Yin, Dr. CHAN Wai Man Johnny, Ms. CHAN Yee Shun Catherine, Dr. CHAN Kwok Keung, Dr. CHOW Kai Ming, Ms. CHU Lai Ming Pauline, Dr. HO Pak Cheong, Dr. HUI Kin Leung Edward, Prof. JOYNT Gavin, Prof. LAI Wai Man Raymond, Mr. LAW Kwok Ming, Dr. LEE Ka Fai, Dr. LEUNG Yin Yan Jenny, Ms. LIU Hor Ki Angela, Dr. MOK Wing Yuk, Ms. NG Vivien, Dr. QUE Tak Lun, Dr. TSANG Tak Yin Owen, Dr. TSE Wing Sze Cindy, Dr. WONG Chi Sing Frank, Ms. WONG Ka Wah Jennifer, Mr. YICK Pak Kin, Prof. ZUO Joan.

## Author Contributions

**Conceptualization:** Ka Lun Alan Wu.

**Formal analysis:** Ka Ho Matthew Hui.

**Investigation:** Chung Yan Grace Lui, Jason Chen, Tai Ning Teddy Lam.

**Methodology:** Ka Ho Matthew Hui, Ka Lun Alan Wu, Yin Ting Cheung, Tai Ning Teddy Lam.

**Project administration:** Ka Ho Matthew Hui, Chung Yan Grace Lui, Yin Ting Cheung.

**Resources:** Chung Yan Grace Lui, Ka Lun Alan Wu, Jason Chen.

**Software:** Ka Ho Matthew Hui.

**Supervision:** Chung Yan Grace Lui, Yin Ting Cheung, Tai Ning Teddy Lam.

**Validation:** Ka Ho Matthew Hui.

**Writing – original draft:** Ka Ho Matthew Hui.

**Writing – review & editing:** Ka Ho Matthew Hui, Chung Yan Grace Lui, Ka Lun Alan Wu, Jason Chen, Yin Ting Cheung.

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
