## [Decision Letter · Decision Letter 0]

5 Jan 2022

PONE-D-21-24164

Multi-center prospective population pharmacokinetic study and the performance of web-based individual dose optimization application of intravenous vancomycin for adults in Hong Kong: A study protocol

PLOS ONE

Dear Dr. Lam,

Thank you for submitting your manuscript to PLOS ONE. After careful consideration, we feel that it has merit but does not fully meet PLOS ONE’s publication criteria as it currently stands. Therefore, we invite you to submit a revised version of the manuscript that addresses the points raised during the review process.

This is an interesting approach that has merit, but the reviewers include several suggestions for improvement.  In particular, please minimize the use of acronyms which often serve only to irritate the reader - we have more than enough 'electron space' to simply spell most of these out save those in the common vernacular (e.g. ICU, KDIGO, etc.).  I look forward to seeing the revision which I will send back to the same referees for decision.

We look forward to receiving your revised manuscript.

Kind regards,

Steven Eric Wolf, MD

Academic Editor

PLOS ONE

Journal Requirements:

3. Your abstract cannot contain citations. Please only include citations in the body text of the manuscript, and ensure that they remain in ascending numerical order on first mention.

Reviewers' comments:

Reviewer's Responses to Questions

**Comments to the Author**

1. Does the manuscript provide a valid rationale for the proposed study, with clearly identified and justified research questions?

Reviewer #1: Yes

Reviewer #2: Partly

Reviewer #3: Yes

2. Is the protocol technically sound and planned in a manner that will lead to a meaningful outcome and allow testing the stated hypotheses?

Reviewer #1: Partly

Reviewer #2: Partly

Reviewer #3: Yes

3. Is the methodology feasible and described in sufficient detail to allow the work to be replicable?

Reviewer #1: Yes

Reviewer #2: Yes

Reviewer #3: Yes

4. Have the authors described where all data underlying the findings will be made available when the study is complete?

Reviewer #1: Yes

Reviewer #2: Yes

Reviewer #3: Yes

5. Is the manuscript presented in an intelligible fashion and written in standard English?

Reviewer #1: Yes

Reviewer #2: Yes

Reviewer #3: Yes

6. Review Comments to the Author

You may also provide optional suggestions and comments to authors that they might find helpful in planning their study.

Reviewer #1: This is an interesting PK study that primarily describes a carefully planned PK analysis. The statistician is clearly an expert in NONMEM7. Although I am a statistical reviewer, I am more concerned that the paper is primarily written about the statistical procedures rather than the design of the study itself. Some of the omissions that I see are:

--multi-center studies should describe how each center's personnel will be trained on the protocol so that patient procedures are homogeneous

--A lot of data will be collected, and this is well-described, but its relation to the study objectives is less clearly described.

--There is no sample size justification. Why 300? Why 50?

--Bayesian methods seem appropriate. Yet you are using t-tests and standard survival estimators for the comparison.

--More on historical controls. How can we be certain that the historical controls are comparable?

--Far too many acronyms throughout the paper. I know PK people like acronyms, but it is overwhelming for the reader to keep track.

--Details such as in lines 208-212 would be better shown in a figure or flow chart.

--Remove the software used in the abstract. NONMEM is not a statistical procedure, it is a software designed to do certain modeling. It is better to say the statistical modeling technique rather than refer to the software.

--A sweeping conclusion that related all these NONMEM C_s analyses to clinical practice.

Reviewer #2: The introduction needs to clarify the other recommendations and guidelines being used to guide the new study.

Recommend addressing that reference method or “gold standard” for calculating MIC is called broth microdilution (BMD) and how that incorporates to the study.

Target range should be AUC:MIC ratio of 400-600.

Reviewer #3: Multi-center prospective pharmacokinetic study and the performance of web-based individual dose optimization application of intravenous vancomycin for adults in Hong Kong: A study protocol

I commend the authors on this ambitious study to develop an algorithm for area under the curve based individual vancomycin dosing.

Line 111: Please define “HA”

Line 137: explain methods for a priori sample sizes

Table 1

Constant Data:

Consider staging acute kidney injury at baseline and throughout the treatment protocol (ie RIFLE, AKIN, or KDIGO). This can easily be calculated with proposed laboratory values to be collected and can further stratify patients adding greater depth and applicability to the web-based model.

Also, include source of infection

Dynamic Data Items:

Include renal based outcome (ie time to development of AKI, Time to ESRD, time to change in AKI stage, etc.)

Line 127: For patients with a new IDCP, please explain how this new IDCP will be treated during the data analysis. Will this be a unique dataset? Define the timeframe between cessation of vancomycin and a new IDCP.

Line 189: Define dosing interval.

Line 227: 168 hours is between 28 and 42 half-lives of vancomycin. This seems excessive. Consider reducing this time frame or explain why this time frame is chosen.

Line 290: Define TTT, TTA, TTN

7. PLOS authors have the option to publish the peer review history of their article (what does this mean?). If published, this will include your full peer review and any attached files.

Reviewer #1: No

Reviewer #2: No

Reviewer #3: No

---

## [Author Response · Author response to Decision Letter 0]

15 Mar 2022

Dear Editor,

All authors and I would like to thank the editorial staff and reviewers for handling our manuscript. Please see the following point-to-point responses to the editors’ and reviewers’ comments. Editors’ and reviewers’ comments are quoted in smaller grey font with tight line spacing, while our responses are in blue.

Journal Requirements:

Thank you for the information. We have reformatted our manuscript accordingly.

The protocol does not report any result. We have specified so in the Data Availability section.

3. Your abstract cannot contain citations. Please only include citations in the body text of the manuscript, and ensure that they remain in ascending numerical order on first mention.

Our abstract does not contain any citation.

We have added the caption for our Supporting Information files at the end of our manuscript and updated any in-text citations accordingly.

Reviewer #1: This is an interesting PK study that primarily describes a carefully planned PK analysis. The statistician is clearly an expert in NONMEM7. Although I am a statistical reviewer, I am more concerned that the paper is primarily written about the statistical procedures rather than the design of the study itself.

Thank you for your time reviewing our manuscript and your valuable comments. We understand that our current manuscript is lacking certain details regarding study design. Please see our responses below.

Some of the omissions that I see are:

--multi-center studies should describe how each center's personnel will be trained on the protocol so that patient procedures are homogeneous

We will explain the protocol thoroughly to the site coordinators, answer all questions they have before study commencement, and closely monitor the recruitment of and study procedures done on the first subject at each center. The section “Compliance to ethical standard, registration, and approval” has been supplemented.

--A lot of data will be collected, and this is well-described, but its relation to the study objectives is less clearly described.

Thank you for your reminder. We have supplemented the “Structural and parameter models” and “Covariate model” sections to explain how the data help characterize the concentration-time profiles of IV vancomycin, considering potential covariate effects.

--There is no sample size justification. Why 300? Why 50?

Unlike hypothesis testing, where the minimum sample size required can be approximated based on the expected effect size and the power needed, in a popPK modeling study where the major covariates are also known a priori, there is no established way to calculate the minimum sample size. The sample sizes of previously published popPK models vary significantly, for e.g., 16 and 36 in Hui K. H. et al (2019) DOI: 10.1002/jcph.1349, 326 in Francis J. et al (2019) DOI: 10.1128/AAC.01964-18, and 1,151 subjects in Stringer F. et al (2013) DOI: 10.1177/0091270012447121. We notice that most popPK reports had sample sizes that range from 50 to 300. Therefore, we aim at the higher end of the range by defining a sample size of 300 and hope for more accurate estimates while avoiding further increasing the study burden.

Similarly, there is no established way to calculate the minimum sample size for external validation. Our decision to include 50 subjects for external validation was based on the observation that most popPK studies adopt a sample size for external validation that is about 15-30% of the sample size for model estimation. (300 * 17% = 50)

--Bayesian methods seem appropriate. Yet you are using t-tests and standard survival estimators for the comparison.

Bayesian methods are being applied to estimate the individual PK and thus optimal doses. However, successfully verifying the prior distribution to be used in the Bayesian estimation process does not imply that using the developed interface to guide dosing will improve the overall clinical outcome. Therefore, t-tests and Cox regression are proposed to compare the clinical outcomes of the two treatment approaches: with vs without the developed Bayesian application. These are statistical tests commonly used to test for treatment effects in conventional clinical studies.

--More on historical controls. How can we be certain that the historical controls are comparable?

We will verify the comparability through baseline characteristics comparison and, if necessary, propensity score matching. We have supplemented this under the sub-section “Statistical analyses for outcomes” in the Material and methods section.

--Far too many acronyms throughout the paper. I know PK people like acronyms, but it is overwhelming for the reader to keep track.

We apologize for using too many acronyms. We have removed most acronyms for technical terms (such as Cs,ss,trough) and those used sparingly in the updated manuscript.

--Details such as in lines 208-212 would be better shown in a figure or flow chart.

Thank you for your recommendation. We think it is a great idea to trim the passage to improve readability. We have added a figure to explain the ideal and alternative sampling schedules with respect to the dosing schedule.

--Remove the software used in the abstract. NONMEM is not a statistical procedure, it is a software designed to do certain modeling. It is better to say the statistical modeling technique rather than refer to the software.

We have removed NONMEM from the abstract and replaced it with non-linear mixed-effect modeling techniques.

--A sweeping conclusion that related all these NONMEM C_s analyses to clinical practice.

We have added a conclusive sentence at the end of the sub-section “PopPK model estimation and validation” in the Materials and methods section. Thank you for the recommendation.

Reviewer #2: The introduction needs to clarify the other recommendations and guidelines being used to guide the new study.

Thank you for your time reviewing our manuscript. We have further explained (1) the choice of a popPK model as the prior distribution for Bayesian estimation as a recommendation of the guideline.

Recommend addressing that reference method or “gold standard” for calculating MIC is called broth microdilution (BMD) and how that incorporates to the study.

Target range should be AUC:MIC ratio of 400-600.

We have included an explanation in the footnote of Table 1 about the methods used to measure MIC.

Reviewer #3: Multi-center prospective pharmacokinetic study and the performance of web-based individual dose optimization application of intravenous vancomycin for adults in Hong Kong: A study protocol

I commend the authors on this ambitious study to develop an algorithm for area under the curve based individual vancomycin dosing.

Thank you very much for your support of our work. Please see our responses to your valuable comments below.

Line 111: Please define “HA”

HA refers to Hospital Authority, the statutory body governing public hospitals in Hong Kong. The definition has been supplemented at its first appearance in the main text.

Line 137: explain methods for a priori sample sizes

Unlike hypothesis testing, where the minimum sample size required can be approximated based on the expected effect size and the power needed, in a popPK modeling study where the major covariates are also known a priori, there is no established way to calculate the minimum sample size. The sample sizes of previously published popPK models vary significantly, for e.g., 16 and 36 in Hui K. H. et al (2019) DOI: 10.1002/jcph.1349, 326 in Francis J. et al (2019) DOI: 10.1128/AAC.01964-18, and 1,151 subjects in Stringer F. et al (2013) DOI: 10.1177/0091270012447121. We notice that most popPK reports had sample sizes that range from 50 to 300. Therefore, we aim at the higher end of the range by defining a sample size of 300 and hope for more accurate estimates while avoiding further increasing the study burden.

Similarly, there is no established way to calculate the minimum sample size for external validation. Our decision to include 50 subjects for external validation was based on the observation that most popPK studies adopt a sample size for external validation that is about 15-30% of the sample size for model estimation. (300 * 17% = 50)

Table 1

Constant Data:

Consider staging acute kidney injury at baseline and throughout the treatment protocol (ie RIFLE, AKIN, or KDIGO). This can easily be calculated with proposed laboratory values to be collected and can further stratify patients adding greater depth and applicability to the web-based model.

Also, include source of infection

Dynamic Data Items:

Include renal based outcome (ie time to development of AKI, Time to ESRD, time to change in AKI stage, etc.)

We agree that adding AKI staging enables deeper analyses, such as the dose relationship of AKI. We have added a statement about conversion to AKI stages at the end of the sub-section “Data exclusion and management” and a description of corresponding statistical analyses in the sub-section “Statistical analyses for outcomes” under the Materials and methods section. Besides, we have added the item “source of infection” in Table 1. Similarly, we have included renal outcomes in terms of AKI staging in Table 1 and the sub-section “Statistical analyses for outcomes”.

Line 127: For patients with a new IDCP, please explain how this new IDCP will be treated during the data analysis. Will this be a unique dataset? Define the timeframe between cessation of vancomycin and a new IDCP.

The primary purpose of defining the IDCP is to ensure the necessary rich Cs sampling near the start of vancomycin treatment. The new IDCP will still belong to the same subject in the dataset. There will be no wash-out period between consecutive IDCPs – once a new IDCP is indicated to start due to vancomycin re-initiation, the remaining vancomycin estimated to be in the subject’s body from the doses in the previous IDCP will be carried over for the estimation of Cs in the subsequent IDCP(s). A sentence has been added to the end of the first paragraph of the sub-section ”Data exclusion and management” under the Materials and methods section.

Line 189: Define dosing interval.

Dosing interval has been defined later in the same paragraph. We apologize for putting the definitions after usage, thus causing confusion. We have moved the definition to the start of the paragraph.

Line 227: 168 hours is between 28 and 42 half-lives of vancomycin. This seems excessive. Consider reducing this time frame or explain why this time frame is chosen.

Thank you for pointing this out. The intention of introducing this time frame is to remove any Cs whose prediction might be inaccurate due to missing dosing records. (E.g. it is reasonable to suspect that there are missing dosing records when vancomycin remains detectable in a subject with normal renal function 1 week after the last dose.) It is true that 168 hours is typically 28 to 42 half-lives of vancomycin. However, this ratio is much reduced in subjects with profound renal impairment. Therefore, we would like to take a 7-day period as an approximated average. Besides, we will keep an eye on any significant underestimation for Cs recorded after a suspiciously long period after the last dose in case any missing dosing records are not being picked up by the above 7-day method.

Line 290: Define TTT, TTA, TTN

To reduce the use of acronyms, as suggested by other reviewers, we have removed the acronyms TTT (time to therapeutic target attainment), TTA (time to afebrile), and TTN (time to normal white blood cell count) from the text.

---

## [Decision Letter · Decision Letter 1]

19 Apr 2022

Multi-center prospective population pharmacokinetic study and the performance of web-based individual dose optimization application of intravenous vancomycin for adults in Hong Kong: A study protocol

PONE-D-21-24164R1

Dear Dr. Lam,

We’re pleased to inform you that your manuscript has been judged scientifically suitable for publication and will be formally accepted for publication once it meets all outstanding technical requirements.

Kind regards,

Steven Eric Wolf, MD

Academic Editor

PLOS ONE

Additional Editor Comments (optional):

Reviewers' comments:

Reviewer's Responses to Questions

**Comments to the Author**

1. Does the manuscript provide a valid rationale for the proposed study, with clearly identified and justified research questions?

Reviewer #1: Yes

2. Is the protocol technically sound and planned in a manner that will lead to a meaningful outcome and allow testing the stated hypotheses?

Reviewer #1: Yes

3. Is the methodology feasible and described in sufficient detail to allow the work to be replicable?

Reviewer #1: Yes

4. Have the authors described where all data underlying the findings will be made available when the study is complete?

Reviewer #1: Yes

5. Is the manuscript presented in an intelligible fashion and written in standard English?

Reviewer #1: Yes

6. Review Comments to the Author

You may also provide optional suggestions and comments to authors that they might find helpful in planning their study.

Reviewer #1: 

7. PLOS authors have the option to publish the peer review history of their article (what does this mean?). If published, this will include your full peer review and any attached files.

Reviewer #1: No

---

## [Editor Report · Acceptance letter]

25 Apr 2022

PONE-D-21-24164R1 

Multi-center prospective population pharmacokinetic study and the performance of web-based individual dose optimization application of intravenous vancomycin for adults in Hong Kong: A study protocol 

Dear Dr. Lam:

I'm pleased to inform you that your manuscript has been deemed suitable for publication in PLOS ONE. Congratulations! Your manuscript is now with our production department. 

Kind regards, 

on behalf of

Dr. Steven Eric Wolf 

Academic Editor

PLOS ONE